# Fibroblast Growth Factor 21 Has a Diverse Role in Energetic and Reproductive Physiological Functions of Female Beef Cattle

**DOI:** 10.3390/ani13203185

**Published:** 2023-10-12

**Authors:** Ligia D. Prezotto, Jessica A. Keane, Andrea S. Cupp, Jennifer F. Thorson

**Affiliations:** 1Department of Animal Science, University of Nebraska-Lincoln, 3940 Fair Street, Lincoln, NE 68583-0908, USA; lprezotto@unl.edu (L.D.P.); jakeane@vt.edu (J.A.K.); acupp2@unl.edu (A.S.C.); 2U.S. Meat Animal Research Center, Agricultural Research Service, United States Department of Agriculture, Clay Center, NE 68933-0166, USA

**Keywords:** bovine, fibroblast growth factor 21, lactation, metabolites, parturition, puberty

## Abstract

**Simple Summary:**

Fibroblast growth factor 21 (FGF21) has been identified in multiple mammalian species as a marker of energetic stress that has the potential to negatively impact nutritional and reproductive performance. However, the role of FGF21 in regulating energetic and reproductive physiology of cattle selected for meat production has yet to be characterized. Within this report, we have conducted a comprehensive evaluation of the role FGF21 plays in female beef cattle throughout the production process. We have revealed that FGF21 plays an expansive role in beef cattle during various developmental and physiological processes. Moreover, this is the only work that has investigated the role of FGF21 in pubertal development, mammary secretions, or postpartum return to reproductive cycles in beef cattle—all other reports have been performed in dairy cattle breeds. In conclusion, FGF21 plays a role in physiological functions in beef cattle that can be applied to advance the understanding of basic scientific processes governing the nutritional regulation of reproductive function but also provides a novel means for beef cattle producers to select parameters of financial interest.

**Abstract:**

Fibroblast growth factor 21 (FGF21) has been identified in multiple mammalian species as a molecular marker of energy metabolism while also providing negative feedback to the gonads. However, the role of FGF21 in regulating the energetic and reproductive physiology of beef heifers and cows has yet to be characterized. Herein, we investigated the temporal concentrations of FGF21 in female beef cattle from the prepubertal period to early lactation. Circulating concentrations of FGF21, non-esterified fatty acids, plasma urea nitrogen, glucose, and progesterone were assessed. Ultrasonography was employed to determine the onset of puberty and resumption of postpartum ovarian cyclicity as well as to measure backfat thickness. Finally, cows and calves underwent the weigh-suckle-weigh technique to estimate rate of milk production. We have revealed that FGF21 has an expansive role in the physiology of female beef cattle, including pubertal onset, adaptation to nutritional transition, rate of body weight gain, circulating markers of metabolism, and rate of milk production. In conclusion, FGF21 plays a role in physiological functions in beef cattle that can be applied to advance the understanding of basic scientific processes governing the nutritional regulation of reproductive function but also provides a novel means for beef cattle producers to select parameters of financial interest.

## 1. Introduction

Changes in feed quality and availability alter the nutritional and physiological status of animals [1]. This change is particularly evident in beef cattle on pastures that must respond metabolically as forage quality and quantity vary by season. Therefore, it is important to better understand the impact of these changes in energy use at different stages of the beef production process. One marker of the metabolic state, fibroblast growth hormone 21 (**FGF21**), has been shown to regulate energy metabolism as well as reproductive functions [2].

Fibroblast growth factor 21 is synthesized and secreted by the liver in response to animal nutritional intake [3] and the stage of production in dairy cattle during late gestation and early lactation [4], with a marked periparturient increase in the circulating concentration and hepatic expression of FGF21 [5]. Kahn et al. [4] proposed that FGF21 is a biomarker for postpartum metabolic stress in dairy cows coordinating homeorhetic adaptations. Circulating concentrations of FGF21 have been correlated with circulating concentrations of glucose, insulin, insulin-like growth factor 1, thyroid hormone, growth hormone, and other metabolic markers by others [4,5,6]; however, no work has evaluated the relationship between FGF21 and parturition or other metabolites in periparturient ruminants selected for beef production.

In addition to the role that FGF21 plays in metabolic regulation, FGF21 has also been demonstrated to regulate reproductive function via nutritionally gated mechanisms in mammalian species. Owen et al. [2] demonstrated that female mice overexpressing FGF21 exhibit abnormal estrous cycles driven by insufficient secretion of LH during proestrus, thus preventing ovulation; however, the selective elimination of the FGF21 co-receptor β-Klotho in the forebrain of female mice rescues this phenotype. Of note, the role of FGF21 can also induce a long-term reduction in fertility [7], which is of particular relevance to beef production practices that rely heavily on retention of reproductively productive females. Therefore, research is warranted to identify the role of circulating FGF21 in the pubertal transition and postpartum resumption of ovarian cyclicity in beef cattle.

In the present study, we aimed to investigate the temporal associations between elevated concentrations of FGF21 relative to physiologic state, concentrations of metabolites, backfat thickness (BF), rate of body weight gain, and degree of milk production in beef cattle. The pronounced rise in the concentration of FGF21 during the periparturient period has been previously characterized in dairy cattle [4,5]. However, no studies have characterized the role of FGF21 in cattle selected for meat production, which preferentially utilize divergent metabolic pathways, nor has the relationship between FGF21 and parameters of production performance (average daily gain, backfat thickness, pubertal attainment, or postpartum resumption of ovarian cyclicity) been evaluated in cattle raised for the production of beef. Based on previous research, we have developed the hypotheses that (1) a decrease in concentration of FGF21 occurs prior to the onset of puberty; (2) the concentration of FGF21 is maintained during the period of transition from a pasture to a dry lot setting; (3) concentrations of FGF21 differ by pubertal status and between phases of the estrous cycle; (4) the concentration of FGF21 is predictive of measures of heifer performance; (5) alterations in circulating markers of metabolism precede changes in the concentration of FGF21 during the periparturient and lactation periods; (6) a decrease in the concentration of FGF21 occurs prior to the postpartum resumption of reproductive cyclicity; and (7) the concentration of FGF21 is predictive of rate of milk production.

## 2. Materials and Methods

Animal care and use protocols were approved by the University of Nebraska-Lincoln Institutional Animal Care and Use Committee (approved outline 881) and the Montana State University Agricultural Animal Care and Use Committee (approved outlines 2014-AA06M and 2015-AA02M).

### 2.1. Animals and Experimental Design

#### 2.1.1. Experiment 1

The objective of this experiment was to characterize the secretion of FGF21 during the peripubertal period in beef heifers. To achieve this objective, heifers comprised 75% Red Angus and 25% MARC III composites (25% Red Angus, 25% Hereford, 25% Pinzgauer, and 25% Red Poll; *n* = 6; 12.8 ± 0.04 months of age; 320.1 ± 5.2 kg at pubertal onset) from the University of Nebraska-Lincoln Physiology research herd were utilized. Heifers were weaned from their dams at approximately 6 months of age and group-housed on a pasture for the duration of the experiment. Heifers were provided additional forage to meet or exceed NRC recommendations [8]. Water and vitamin/mineral supplements were offered *ad libitum*. Blood samples were collected weekly via caudal venipuncture for harvest of plasma. Plasma samples were analyzed for concentrations of progesterone and FGF21. Circulating concentrations of progesterone were utilized to identify reproductive status of heifers (prepubertal vs. pubertal). Heifers were classified as pubertal once serum concentrations of progesterone were ≥1.0 ng/mL from blood harvested weekly and continuation of reproductive cyclicity was confirmed by half of the subsequent weekly blood samples having serum concentrations of progesterone ≥1.0 ng/mL. Circulating concentrations of FGF21 were standardized relative to the onset of puberty (*n* = 6 heifers), then evaluated using ANOVA with repeated measures (day) with day as the source of variation and animal as the subject. The least-squares means procedure was used to compare means once significant differences were detected. Data are reported as least-squares means +SEM. Significance was declared at *p* ≤ 0.05.

#### 2.1.2. Experiment 2

The objectives of this experiment were to determine if (1) serum concentrations of FGF21 differ relative to entry into a dry lot, (2) serum concentrations of FGF21 differ with reproductive status, and (3) the serum concentration of FGF21 is predictive of backfat thickness (**BF**) and average daily gain (**ADG**) in beef heifers. To achieve these objectives, Black Angus heifers (*n* = 33; 8.0 ± 0.1 months of age; 273.4 ± 24.0 kg at onset of experiment) were utilized. Heifers were removed from the pasture and housed in a dry lot fitted with electronic gates (GrowSafe Systems Ltd., Airdrie, AB, Canada) where they were offered a total mixed ration to meet or exceed NRC recommendations [9] for the 63-day feeding period. Water and vitamin/mineral supplements were offered *ad libitum*. Body weight was determined on 2 consecutive days at the beginning, middle, and end of the feeding period, and the feed was adjusted at mid-test to maintain a targeted ADG of 0.45 kg of BW/d. Heifers had feed withdrawn for 16 h to collect body weight and blood samples via jugular venipuncture for harvest of serum. Serum samples were analyzed for concentrations of FGF21 and progesterone. To confirm the reproductive status of heifers (prepubertal, pubertal during the follicular phase of the estrous cycle, or pubertal within the luteal phase of the estrous cycle), transrectal ultrasonography was also employed on experimental day 56. Transrectal ultrasonography was employed to visualize ovarian structures (follicles, corpora lutea, and corpora albicans) and endometrial edema. Prepubertal heifers (*n* = 10) had no corpora lutea, corpora albicans, or follicles greater than 10 mm present on the ovary and serum concentrations of progesterone <1 ng/mL on days −7, 0, 28, and 56. Heifers classified as pubertal were separated by phase of the estrous cycle (follicular or luteal phase of the estrous cycle). Pubertal heifers within the follicular phase of the estrous cycle (*n* = 9) exhibited an ovarian follicle greater than 10 mm, endometrial edema, serum concentrations of progesterone <1 ng/mL on day 56, and serum concentrations of progesterone ≥1 ng/mL on days −7, 0, or 28. Pubertal heifers within the luteal phase of the estrous cycle (*n* = 11) exhibited an ovarian corpus luteum, lacked endometrial edema, had no ovarian follicles greater than 10 mm, and serum concentrations of progesterone ≥1 ng/mL on day 56. To determine the nutritional performance of heifers, individual ADG and BF were determined. Individual ADG was calculated for the 56-day feeding period. On day 56, ultrasonic measurements of BF were taken between the 12th and 13th ribs. Images were collected using a SonoSite Edge II imaging system equipped with a 10-5 MHz (SonoSite, Inc.; Bothell, WA, USA), 15 cm linear array transducer. Backfat thickness was measured within captured images using integrated software with distance calipers. The effect of day of the experiment on concentrations of FGF21 and progesterone were evaluated using ANOVA with repeated measures (day), and day was used as the repeated variable and animal used as the subject. The effect of the reproductive status on concentrations of FGF21 and progesterone were evaluated using ANOVA, with the sources of variation being reproductive status. The least-squares means procedure was used to compare means once significant differences were detected. Data are reported as least-squares means ±SEM. Pearson correlation coefficients were determined using PROC CORR, whereas linear regression analysis was performed using PROC REG to relate FGF21 on day 56 of the experiment to BF on day 56 of the experiment and ADG over the course of the 56-day experiment. Significance was declared at *p* ≤ 0.05.

#### 2.1.3. Experiment 3

The objective of this experiment was to characterize the secretion of FGF21 and metabolites during the periparturient period and early lactation in beef cows. To achieve this objective, pregnant, multiparous Black Angus cows (*n* = 30; 4 to 5 years of age; 608.4 ± 40.4 kg at onset of experiment) were fed a total mixed ration to meet or exceed NRC recommendations [9]. Water and vitamin/mineral supplements were offered *ad libitum*. Feed was withdrawn from cows for 16 h to record body weights, and blood samples were collected via jugular venipuncture on days −14, −7, 0 (within 12 h of calving), 14, 28, and 60, relative to parturition for the harvest of serum and plasma. The concentration of FGF21 was assessed in the serum, while concentrations of non-esterified fatty acids (**NEFA**), plasma urea nitrogen (**PUN**), and glucose were assessed in the plasma. Circulating concentrations of FGF21, NEFA, PUN, and glucose were evaluated using ANOVA with repeated measures (day) with day relative to parturition, age of cow, sex of calf, length of gestation, and day by cow age interaction as the source of variation and animal as the subject. The least-squares means procedure was used to compare means once significant differences were detected. Data are reported as least-squares means ±SEM. Significance was declared at *p* ≤ 0.05.

#### 2.1.4. Experiment 4

The objectives of this experiment were to determine if (1) serum concentrations of FGF21 fall prior to the first postpartum ovulation and if (2) serum concentrations of FGF21 differ by level of milk production in beef cows. To achieve these objectives, pregnant, multiparous Black Angus cows (*n* = 50; 3 to 6 years of age; 586.2 ± 50.3 kg at onset of experiment) were fed a total mixed ration to meet or exceed NRC recommendations [9]. Water and vitamin/mineral supplements were offered *ad libitum*. Cows had feed withheld for 16 h to record body weights, and blood samples were collected via jugular venipuncture on days 0 (within 24 h of calving), 7, 14, 21, 28, 32, 36, 40, 44, 48, 52, 56, 60, 64, 68, and 72, relative to parturition for harvest of serum and analysis of concentrations of FGF21 and progesterone. Circulating concentrations of progesterone were utilized to identify the reproductive status of cows (cyclic vs. noncyclic). Cows were classified as cyclic once two consecutive serum samples had concentrations of progesterone ≥1.0 ng/mL. Cows that failed to cycle during the 72-day experiment were assigned to have begun cycling on day 72 for statistical analysis. Cows were also aligned relative to the day of first postpartum ovulation to determine differences in circulating concentrations of FGF21 by day and influence of the sex of the suckling calf. On day 60, calves were isolated from their dams for 8 h, and the weigh-suckle-weigh procedure [10] was performed in groups of 5 to 7 calves to estimate daily milk production of the dam. Individual ADG was ranked relative to 0.5 standard deviations above and below the targeted ADG (0.00 kg per day; Low (<−0.28 kg per day; *n* = 7), Mid (−0.28 to 0.28 kg per day; *n* = 21), or High (>0.28 kg per day; *n* = 19)) for statistical analysis. Individual daily milk production was ranked relative to 0.5 standard deviations above and below the mean daily milk production (15.04 kg; Low (<10.59 kg; *n* = 15), Mid (10.59 to 19.48 kg; *n* = 18), or High (>19.48 kg; *n* = 13)) for statistical analysis. Circulating concentrations of FGF21 were evaluated using ANOVA with repeated measures (day) with day relative to parturition, age of cow, and day by cow age interaction as the sources of variation and animal as the subject. Circulating concentrations of FGF21 were also evaluated using ANOVA with repeated measures (day) with day relative to first postpartum ovulation, sex of calf, age of cow, and day by cow age interaction as the sources of variation and animal as the subject. Differences between daily milk production classifications were subjected to ANOVA using the GLM procedure of SAS with the age of cow, ADG, and sex of the calf as the fixed effects. The least-squares means procedure was used to compare means once significant differences were detected. Data are reported as least-squares means ±SEM, unless reported otherwise. Significance was declared at *p* ≤ 0.05.

### 2.2. Harvest of Serum and Plasma

Blood samples were collected in evacuated tubes containing no additives or EDTA for the collection of serum or plasma, respectively. Tubes were placed immediately on ice for 30 min. Evacuated tubes containing no additives were allowed to clot at room temperature for 60 min. Samples were centrifuged at 2000× *g* for 15 min, serum and plasma were harvested, and samples were stored at −20 °C until analysis.

### 2.3. Hormone and Metabolite Assays

Concentrations of progesterone for Experiment 1 were determined utilizing a commercially available RIA kit (207270; MP Biomedicals; Irvine, CA, USA) according to manufacturer’s recommendations. Intra-assay coefficients of variation averaged 6.39% and inter-assay coefficients of variation were 7.25% for the analysis of samples containing 0.5, 1.0, and 5.0 ng progesterone/mL. Concentrations of FGF21 (RD291108200R; BioVendor, LLC; Asheville, NC, USA), progesterone (Experiment 2; 4825; Monobind Inc.; Lake Forest, CA, USA), NEFA (999-34691; Wako Diagnostics; Terra Bella, CA, USA), PUN (MAK006; Sigma-Aldrich; St. Louis, MO, USA), and glucose (TR15421; Thermo Fisher Scientific; Middletown, VA, USA) were determined utilizing commercially available ELISA kits according to manufacturer’s recommendations. Intra-assay coefficients of variation averaged 3.64, 5.71, 5.26, 1.42, and 5.21% and inter-assay coefficients of variation were 7.53, 14.69, 6.49, 4.21, and 5.95% for analysis of samples containing 25.6 pmol FGF21/L, 0.64 ng progesterone/mL, 0.67 mmol NEFA/L, 4.8 mmol PUN/L, and 9.3 mmol glucose/L, respectively. Recovery of the added mass of 8 replicates each containing 100 pg of recombinant bovine FGF21 (cyt-657; Prospec; Ness Ziona, Israel) resulted in 115.65% recovery.

## 3. Results

### 3.1. Experiment 1

Results for this experiment are depicted in Figure 1. Concentrations of FGF21 were greater (*p* = 0.05) during the week puberty was attained when compared to 4 weeks prior to the pubertal onset. However, no differences (*p* ≥ 0.07) were observed when comparing concentrations of FGF21 during the week of puberty onset with 1 to 3 weeks prior to the onset of puberty. Moreover, 2 weeks after puberty was attained, concentrations of FGF21 increased again to concentrations greater than those observed during weeks 3 and 4 prior to the onset of puberty (*p* ≤ 0.006) as well as compared to concentrations during the week following the onset of puberty (*p* = 0.03).

### 3.2. Experiment 2

Results for this experiment are depicted in Figure 2, Figure 3 and Figure 4. As expected without implementation of an estrus synchronized protocol, circulating concentrations of progesterone did not differ (*p* = 0.23) by experimental day. Circulating concentrations of FGF21 did differ (*p* < 0.0001) by day of experiment with concentration of FGF21 elevated (*p* < 0.0001) within 7 days of bringing heifers into the dry lot. However, the concentration of FGF21 returned (*p* = 0.29) to values observed on day −7 by day 56 of the experiment.

On day 56 of the experiment, transrectal ultrasonography was employed, and circulating concentrations of progesterone were quantified to classify heifers by reproductive status (prepubertal, pubertal within the follicular phase of the estrous cycle, or pubertal within the luteal phase of the estrous cycle). As expected, circulating concentrations of progesterone on day 56 were greatest (*p* ≤ 0.002) in heifers within the luteal phase of the estrous cycle but did not differ (*p* = 0.89) between heifers classified as prepubertal and pubertal within the follicular phase of the estrous cycle. Intriguingly, circulating concentrations of FGF21 on day 56 did not differ (*p* = 0.09) in heifers by reproductive status.

Finally, markers of heifer nutritional performance (BF on day 56 and ADG over the course of the 56-day feeding period) were plotted against circulating concentrations of FGF21. Interestingly, circulating concentrations of FGF21 on day 56 were predictive of ADG over the course of the 56-day feeding period (*p* = 0.002, y = −0.0046x + 0.5022, and R^2^ = 0.2803) but not predictive of BF on day 56 (*p* = 0.31, y = 0.0003x + 0.1737, and R^2^ = 0.0331).

### 3.3. Experiment 3

The results for this experiment are depicted in Figure 5. The results of this experiment revealed that circulating concentrations of FGF21, NEFA, PUN, and glucose change (*p* < 0.0001) in response to the interval to parturition. Moreover, we have observed that the prepartum reduction in NEFA and PUN precede the periparturient elevation in circulating concentrations of FGF21.

A steady increase (*p* = 0.003) in concentration of FGF21 was observed from day −14 to parturition. Within 14 days of parturition, circulating concentrations of FGF21 returned to values no different (*p* = 0.23) than concentrations recorded 14 days prior to parturition and remained unchanged (*p* ≥ 0.08) for the remainder of the experiment. Ther was no influence of cow age (*p* = 0.72) or cow age by day interaction (*p* = 0.85) on concentrations of FGF21.

Concentrations of NEFA decreased (*p* < 0.0001) from day −14 to day −7, followed by a steady increase (*p* = 0.003) from 7 days prior to parturition to 14 days postpartum, which was maintained (*p* ≥ 0.16) out to day 60. There was no influence of cow age (*p* = 0.20) or cow age by day interaction (*p*= 0.57) on concentrations of NEFA.

Concentrations of PUN decreased (*p* ≤ 0.003) from day −14 to day −7, followed by a rapid increase (*p* < 0.0001) in concentration of PUN from day −7 to day 0. After parturition, concentrations of PUN returned to concentrations no different (*p* ≥ 0.28) than that observed prior to parturition. There was no influence of cow age (*p* = 0.64) or cow age by day interaction (*p* = 0.18) on concentrations of PUN.

Circulating concentrations of glucose were static (*p* = 0.18) during late parturition but increased (*p* < 0.0001) at parturition. However, within 14 days of parturition, concentrations of glucose decreased (*p* = 0.04) then remained the same (*p* ≥ 0.56) for the remainder of the experiment. There was no influence of cow age (*p* = 0.84) or cow age by day interaction (*p* = 0.54) on concentrations of glucose.

### 3.4. Experiment 4

The results for this experiment are depicted in Figure 6, Figure 7 and Figure 8. As also observed in Experiment 3, circulating concentrations of FGF21 were greatest (*p* < 0.0001) at parturition followed by a rapid decrease (*p* ≤ 0.0001) within 7 days of parturition that continued to fall (*p* = 0.001) out to 36 days postpartum. Concentrations of FGF21 began to rise (*p* = 0.02) again by 44 days postpartum and continued to increase (*p* = 0.04) out to 72 days postpartum when peak lactation occurs in beef cattle. However, there was no influence of cow age (*p* = 0.61) or day by cow age interaction (*p* = 0.43) on circulating concentrations of FGF21.

The average interval from parturition to the first ovulation was 51.2 days. However, 20% of cows failed to resume reproductive cyclicity by the end of the experiment and thus were assigned to have resumed reproductive cyclicity at day 72 for the calculation of this interval. When the day was standardized to the day of first postpartum ovulation for animals with confirmed resumption of reproductive cyclicity, concentrations of FGF21 failed to fall (*p* = 0.45) prior to ovulation as hypothesized. However, there was an influence of calf sex, with cows rearing female calves having elevated (*p* = 0.02) concentrations of FGF21 during the period from 14 days before ovulation to 16 days following ovulation when compared to cows rearing male calves (42.2 ± 6.5 pmol FGF21/L and 20.4 ± 6.2 pmol FGF21/L for female and male calves, respectively), but there was no influence of cow age (*p* = 0.40) or cow age by day interaction (*p* = 0.68) during this period.

Upon classifying cows by daily milk production, it was revealed that cows classified as high daily milk producers had greater (*p* ≤ 0.05) concentrations of FGF21 on day 60 when compared to cows classified as having low or moderate daily milk production, but there was no difference (*p* = 0.52) between cows classified as having low or moderate daily milk production. Interestingly, there was no influence of cow age (*p* = 0.34), cow ADG classification (*p* = 0.23), or sex of the calf (*p* = 0.64) on daily milk production classification.

## 4. Discussion

We have conducted a comprehensive evaluation of the role FGF21 plays in female beef cattle throughout the production process. Moreover, we have demonstrated the role that circulating concentrations of FGF21 play in beef cattle during various developmental and physiological processes. To our knowledge, this is the only work that has investigated the role of FGF21 in pubertal development, mammary secretions, or postpartum recrudescence of the reproductive cyclicity of beef cattle—all other reports have been performed in dairy breeds.

### 4.1. Pubertal Transition

The mechanisms integral for the attainment of puberty in cattle are not fully understood, but there is evidence for nutrient-gated mechanisms that regulate pubertal attainment in beef heifers [11,12,13]. Herein, we have expanded the body of work supporting the role of nutrition in the processes required for pubertal development in heifers. In contrast to our hypothesis, we revealed an upward trajectory in circulating concentrations of FGF21 during the prepubertal period. In hindsight we believe this increase in circulating concentrations of FGF21 is potentially a mechanism to increase insulin sensitization and reduce lipolysis of adipose tissue [14] during the peripubertal period, thus allowing for partitioning of steroid hormone precursors toward the production of steroid hormones integral for the attainment of puberty.

### 4.2. Nutritional Transition

While unexpected, heifers transitioning from a pasture to a dry lot setting exhibited elevated circulating concentrations of FGF21. The authors propose that this change was induced by transition to a diet rich in starch. Similar reports have also observed that hepatic FGF21 expression and circulating concentrations of FGF21 rise robustly when diets high in carbohydrates are offered over an acute or prolonged period [15,16,17,18]. Moreover, the current report illustrates that the circulating concentration of FGF21 returns to baseline upon extended exposure to the starch-rich diet as previously reported [19]. While it is unclear what mechanism is driving the protracted elevation in the circulating concentrations of FGF21 in the current report to other reports, the authors propose that it is potentially related to differences in the type of digestive system and age of experimental animals, as well as genetic selection practices utilized in beef production systems toward animals selected for conservation of fat stores.

### 4.3. Heifer Reproductive Status

This study has demonstrated for the first time that FGF21 is not divergent between prepubertal and postpubertal heifers nor across the phases of the estrous cycle of beef heifers. Our finding that the circulating concentration of FGF21 is not correlated with the circulating concentration of progesterone in beef heifers contradicts the findings of others [20] that observed a negative correlation between circulating FGF21 and progesterone in women. Another report [21] also supports these findings [20] indirectly by presenting that the activation of the hepatic estrogen receptor-α increases energy expenditure by stimulating the production of FGF21 in mice. The differences in findings between the aforementioned and current report are potentially attributed to differences in species, divergent regulatory mechanisms of action, or differences in lag time to observable change in circulatory FGF21 between models.

### 4.4. Heifer Nutrient Utilization

The understanding of the role of FGF21 in the accretion of adiposity is incomplete, especially regarding subcutaneous fat stores. It has been demonstrated in an adult, male mouse model that FGF21 signaling directly to adipose tissue is required for the acute insulin-sensitizing effects of FGF21 and, therefore, the reduction in the rate of lipolysis [17]. However, BonDurant et al. [17] also demonstrated that the chronic effects of FGF21 on adipose tissue are not required for changes in energy expenditure or body weight. Kong et al. [22] have recently demonstrated that FGF21 reduces lipid accumulation in hepatocytes harvested from neonatal dairy calves. Moreover, it has been shown that FGF21 drives a decline in the accumulation of triglycerides in muscle of neonatal piglets [23], but measures of subcutaneous adiposity were not assessed in that report. Cumulatively, the aforementioned results support the current findings that FGF21 has the ability to alter body weight, but we are perplexed as to why we were unable to find a correlation between FGF21 and depth of subcutaneous fat over the ribs, given the findings from other species. Potentially, these differences are attributed to the role of FGF21 in subcutaneous fat versus that of intramuscular or hepatic adiposity.

### 4.5. Periparturient Transitional Metabolites

When analyzing the pattern of secretion of FGF21 in periparturient beef cows that were offered a ration to meet or exceed nutrient recommendations, we demonstrated a steady increase in circulating concentrations of FGF21 leading up to the day of parturition, followed by a rapid decline postpartum. Schoenberg et al. [5] have previously demonstrated a similar, yet more abrupt, pattern in periparturient dairy cows that were estimated to have an energy deficit averaging 5 Mcal/d. Interestingly, when Khan et al. [4] provided excess dietary energy to dairy cattle during the periparturient period, a pattern similar to what was observed in the present study was observed—a muted periparturient increase in circulating concentrations of FGF21. Therefore, illustrating that the paramount factor contributing to the periparturient rise in circulating concentrations of FGF21 observed in the bovine is driven by the inability of the dams to consume sufficient dietary energy. However, it must also be noted that the placenta, while not a primary source of circulating FGF21 in rodents [24], may serve as a source of circulating FGF21 in the periparturient period of cattle and thus have the potential to contribute to the periparturient pattern of FGF21 secretion.

To further investigate the metabolic status of beef cows, we analyzed concentration of metabolites (NEFA, PUN, and glucose) from 14 days prior to parturition to 60 days postpartum. Our results contradict that of others [25], which demonstrate that FGF21 induces a reduction in concentrations of NEFA in non-ruminants but is indirectly supported by others [22] who have revealed that FGF21 induces a reduction in the hepatic synthesis of very-low-density lipoproteins in ruminants. This difference may be attributed to species, timeframe samples collected, or physiological state of the animal. Therefore, future research is warranted to determine if energy deficit in periparturient beef cows is regulated via the peroxisome proliferator-activated receptor-α pathway [3,26]. Interestingly, we were able to demonstrate that concentration of NEFA in beef cows fell 7 days prior to parturition, and then slowly increased during lactation. We believe this decline in concentration of NEFA prior to parturition is a function of increasing fetal size, reducing feed intake. This finding aligns with some reports in dairy cattle [27,28] but was not observed by others [29,30]. It is possible that differences in the circulating concentrations of NEFA in response to impending parturition from these reports to the current ones are attributed to breed, nutrient composition of feed, body reserves, as well as time and duration of sampling. Concentration of NEFA slowly increased during lactation in the current report; however, these concentrations never rose to concentrations observed two weeks prior to parturition. Non-esterified fatty acids can be utilized as an alternate source of ATP production through oxidation mechanisms and gluconeogenesis. The increase in NEFA during lactation demonstrates the utilization of NEFA to promote mammary gland secretions directly but also indirectly through esterification within hepatocytes for the eventual production of very-low-density lipoproteins exported to the mammary glands [22,31].

The authors found it critical to also characterize the secretory pattern of PUN during late gestation and early lactation in beef cows, as no previous reports have investigated the relationship between PUN and FGF21. Interestingly, we were able to demonstrate a decrease in concentrations of PUN prior to parturition followed by a second decline during early lactation that followed a transient resurgence at parturition. The authors believe the driving factor for each of these declines is a function of different mechanisms with the reduction in concentrations of PUN prior to parturition, a result of an inability to consume sufficient dietary supply and thus dietary protein; whereas, the lactational decline in concentrations of PUN is attributed to the normalization in dietary intake coupled with the greatest metabolic demand faced by the dam as she transitions into lactation-induced catabolism of muscle tissue for energy production via gluconeogenesis. The transient resurgent rise in PUN at parturition may also be attributed to FGF21-induced muscle atrophy as indirectly supported by others [32] using a rodent model. In dairy cows, others have reported a decrease in metabolized protein from parturition to day 7 of lactation in response to a vast array of metabolic changes [33]. However, this change appears to occur earlier in the beef cow as a reduction in the concentration of PUN occurs prior to parturition. In contrast, the increase in the concentration of PUN at parturition occurs concurrently with the rise in the concentration of FGF21. Interestingly, Hill et al. [34] have recently shown that FGF21 is required for the metabolic adaption to reduce dietary protein intake. From this finding, we believe that the metabolic changes occurring just prior to parturition with an upward trajectory in circulating concentrations of FGF21 may be the factor driving a dietary-induced decline in the circulating concentration of PUN.

Finally, we assessed the influence of day relative to parturition on the circulating concentration of glucose. In agreement with prior reports, the circulating concentration of glucose remained static during late gestation and then exhibited a transient increase at parturition in dairy and beef cows [35,36,37]. The increase in the concentration of FGF21 at parturition, followed by the reduction in the circulating concentration of glucose during lactation, agrees with prior reports [14,38]. Following parturition, circulating concentrations of glucose fell to values below those observed during late gestation, as previously reported in dairy and beef cows [35,36,37,39]. It is proposed that the mechanisms underlying the FGF21-induced reduction in circulating glucose include increased circulating concentrations of insulin coupled with increased cellular expression of the glucose 1 transporter, as reported in other mammalian species [40,41].

### 4.6. Postpartum Resumption of Reproductive Cyclicity

As a result of metabolic changes occurring during the periparturient period, it was plausible to infer that parameters of reproductive performance may be impacted, including the postpartum resumption of reproductive cyclicity in beef cattle. As FGF21 is a marker of metabolic stress, we hypothesized that circulating concentrations of FGF21 would fall prior to the postpartum resumption of reproductive cyclicity. Wang et al. [42] reported that serum concentrations of FGF21 in early lactation are predictive of dairy cows within the follicular phase of the estrous cycle between 50 and 55 days postpartum. Unfortunately, this report failed to identify dairy cows that had ovulated prior to 50 days postpartum and thus failed to differentiate between cows within the luteal phase of the estrous cycle and lactational anovulatory animals. In the current report, we were able to identify beef cow postpartum intervals via circulating concentrations of progesterone throughout early lactation and thus assess the potential contribution of circulating FGF21 on postpartum resumption of reproductive cyclicity. However, we observed that circulating FGF21 is not predictive of the resumption of reproductive cyclicity in beef cattle. Crowe [43] has reported that the interval to the first ovulation in beef cows in a good body condition is approximately 30 days, while the interval to the first ovulation in cows of a poor body condition can take upwards of 100 days. Herein, the interval to the first ovulation averaged 51 days with only 80% ovulating by the end of the 72-day experimental period. From the report by Crowe [43], we can infer that animals within the current experiment were exposed to metabolic stress during lactation. Interestingly, the concentration of FGF21 during lactation in the current experiment was comparable to previous reports in dairy cattle, regardless of energy balance [4,5], and thus illustrates factors beyond body condition and gross energy balance influencing postpartum resumption of the reproductive cyclicity independent of the circulating concentration of FGF21.

### 4.7. Lactational Performance

Lactation is heavily dependent on the use of lipid reserves via integrated metabolic activity between adipose and hepatic tissues. However, the mechanisms behind this integration during lactation are not well understood. Schoenberg et al. [11] have proposed that FGF21 plays a role in coordinating lipid usage during lactation as supported by others in dairy cattle [44]. The mechanism of action that stimulates FGF21 synthesis is mediated by peroxisome proliferator-activated receptor-α that is activated in response to changes in circulating concentrations of NEFA [45], as also observed in the current report. A prior report [5] also supports the current observation that high milk production is accompanied by elevated circulating concentrations of FGF21. This increase in circulating FGF21 may also be the result of lactation-induced energy deprivation driving an increase in hepatic *FGF21* expression, as well as circulating concentrations of FGF21 as previously shown by others in dairy cows [5,45,46,47]. The current findings on milk production are of particular importance to beef cattle production systems as changes in FGF21 are independent of cow age or rate of body weight gain and thus have the potential to aid in beef cow selection to improve lactational performance.

## 5. Conclusions

Within this report we have revealed that FGF21 has an expansive role in the physiology of female beef cattle, including pubertal onset, adaptation to nutritional transition, rate of body weight gain, circulating markers of metabolism, and rate of milk production. Moreover, this is the only work that has investigated the role of FGF21 in pubertal development, mammary secretions, or postpartum recrudescence of the reproductive cyclicity of beef cattle—all other reports have been conducted in dairy breeds. In conclusion, FGF21 plays a role in physiological functions in beef cattle that can be applied to advance the understanding of basic scientific processes governing the nutritional regulation of reproductive function but also provides a novel means for beef cattle producers to select parameters of financial interest.

## Figures and Tables

**Figure 1 animals-13-03185-f001:**
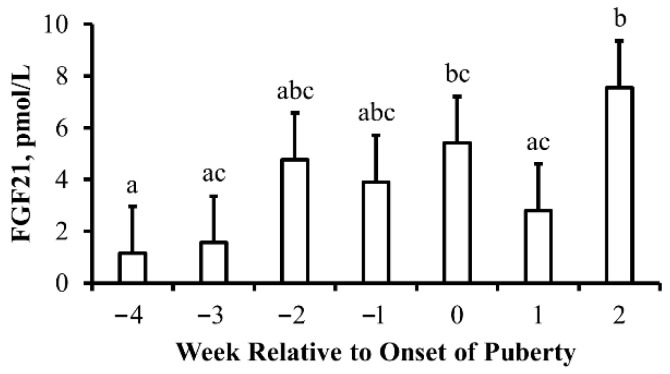
**Characterization of the secretion of fibroblast growth factor (FGF21) during the peripubertal period in beef heifers:** Least-squares means (+SEM) serum concentrations of FGF21 (pmol/L) in beef heifers by week relative to onset of puberty. Means with different lowercase letters denote differences between weeks. Week *p* = 0.05 and *n* = 6 heifers per week.

**Figure 2 animals-13-03185-f002:**
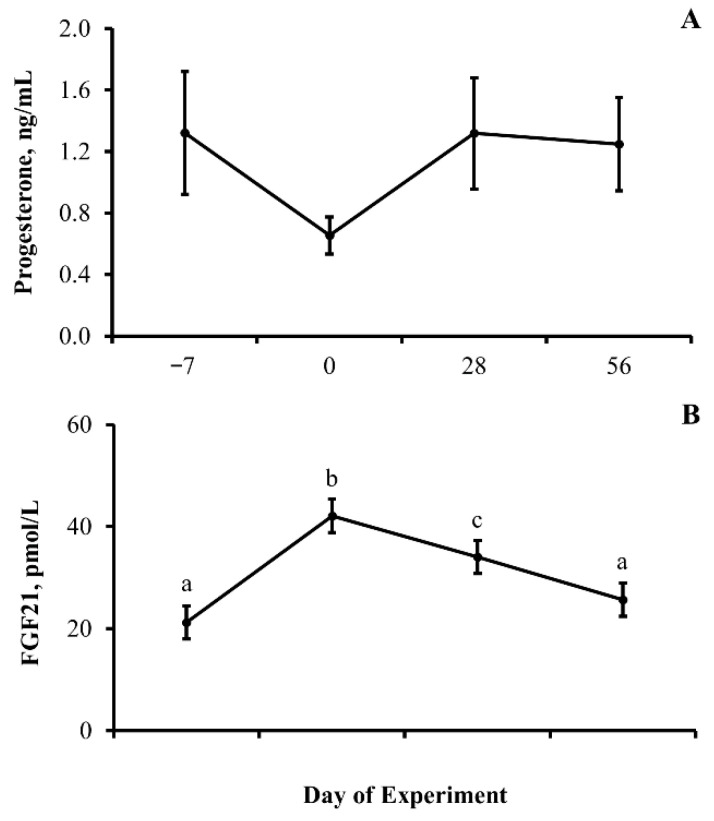
**Serum concentrations of fibroblast growth factor 21 (FGF21) in beef heifers differ upon transition from a pasture to a dry lot setting, independent of reproductive status:** Least-squares means (±SEM) serum concentrations of progesterone (ng/mL; (**A**)) and FGF21 (pmol/L; (**B**)) relative to beef heifers entering a feed lot. Means with different lowercase letters denote differences between days. FGF21 by day *p* < 0.0001; progesterone by day *p* = 0.23; and *n* = 33 heifers per day.

**Figure 3 animals-13-03185-f003:**
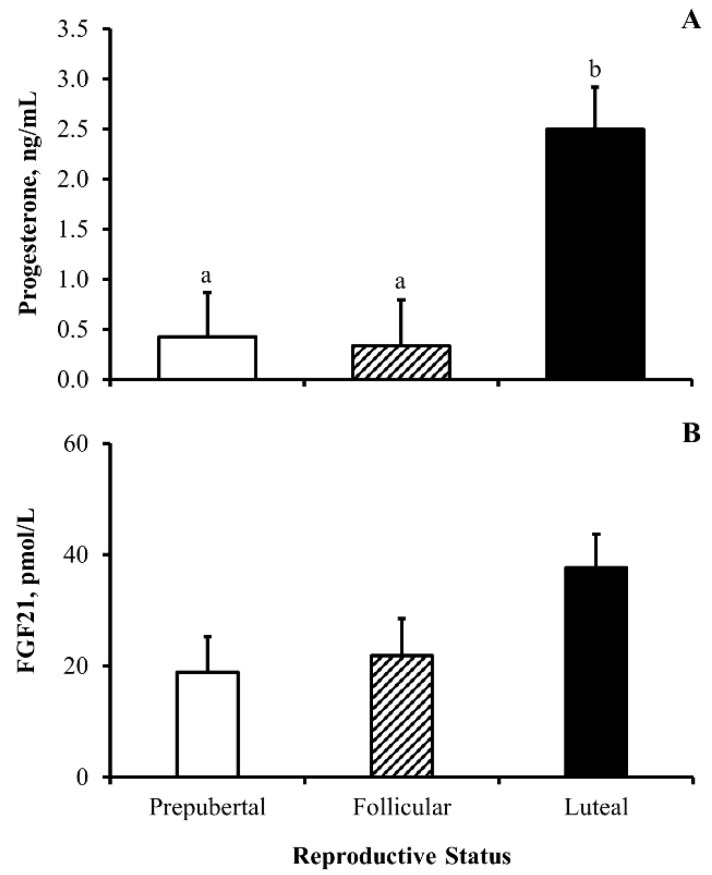
**Serum concentrations of fibroblast growth factor 21 (FGF21) in beef heifers are independent of reproductive status:** Least-squares means (+SEM) serum concentrations of progesterone (ng/mL; (**A**)) and FGF21 (pmo/L; (**B**)) in beef heifers by reproductive status. Means with different lowercase letters denote differences between status. FGF21 by reproductive status *p* = 0.09; progesterone by reproductive status *p* < 0.0001; and *n* = 10, 9, and 11 heifers classified as prepuberal (white bars) or pubertal within the follicular phase of the estrous cycle (follicular; dashed bars) and pubertal within the luteal phase of the estrous cycle (luteal; black bars).

**Figure 4 animals-13-03185-f004:**
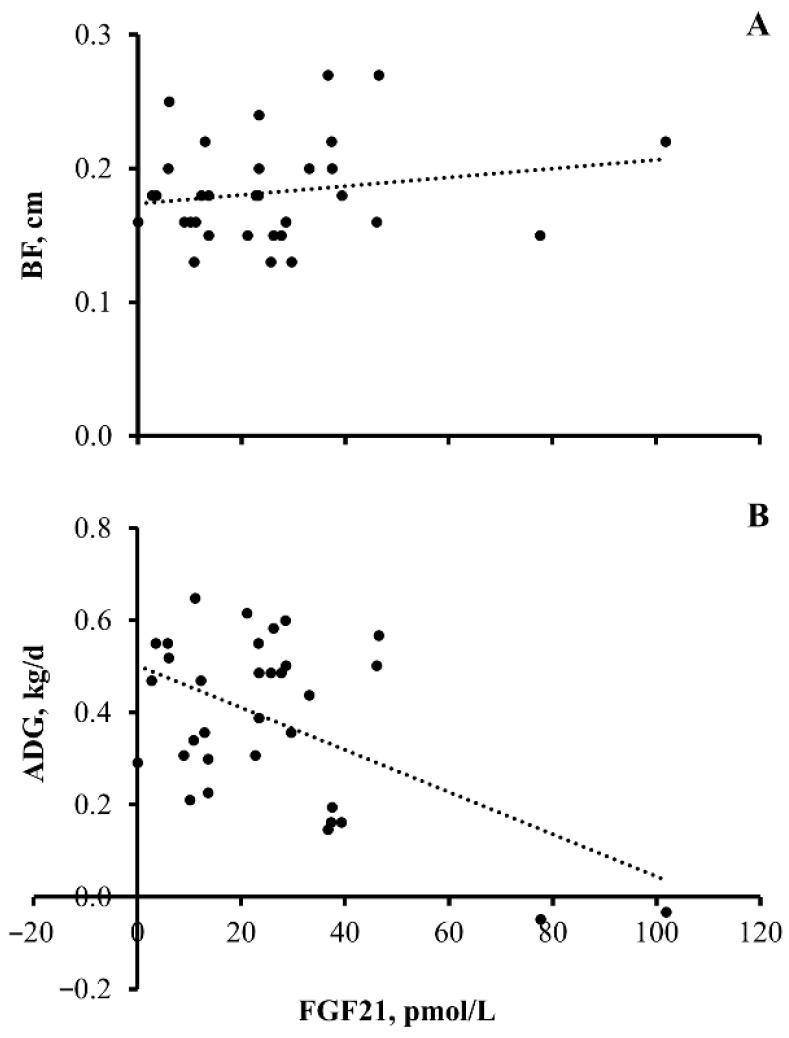
**Serum concentration of fibroblast growth factor 21 (FGF21) is predictive of average daily gain (ADG) but not back fat thickness (BF) in beef heifers:** Plot of BF (cm; (**A**)) on day 56 of the experiment and ADG (kg/d; (**B**)) over the course of the 56-day experiment with FGF21 (pmol/L) on day 56 of the experiment in beef heifers. BF: *p* = 0.31, y = 0.0003x + 0.1737, R^2^ = 0.0331, and *n* = 33 heifers; ADG: *p* = 0.002, y = −0.0046x + 0.5022, R^2^ = 0.2803, and *n* = 33 heifers.

**Figure 5 animals-13-03185-f005:**
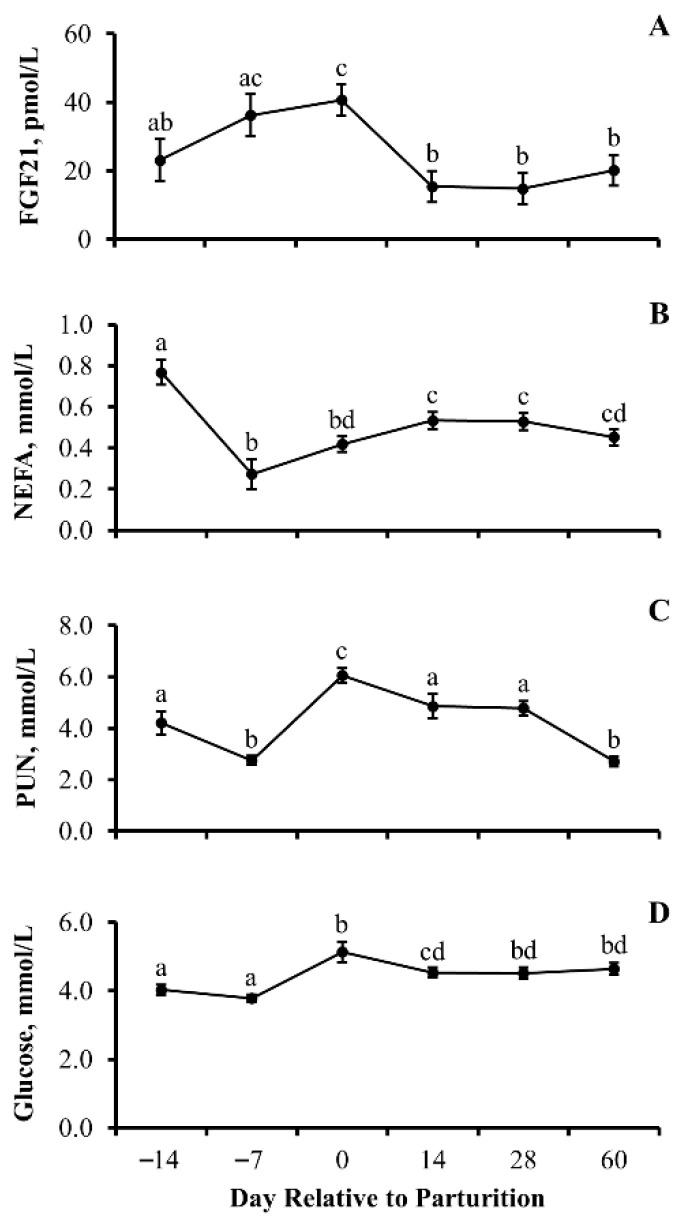
**Prepartum changes in circulating concentrations of non-esterified fatty acids (NEFA) and plasma urea nitrogen (PUN) precede the periparturient elevation in circulating concentrations of fibroblast growth factor 21 in beef cows:** Least-squares means (±SEM) serum concentrations of FGF21 (pmol/L; (**A**)), non-esterified fatty acids (NEFA; mmol/L; (**B**)), plasma urea nitrogen (PUN; mmol/L; (**C**)), and glucose (mmol/L; (**D**)) in beef cows by day relative to parturition. Means with different lowercase letters denote differences between days. FGF21 day *p* < 0.0001, NEFA day *p* < 0.0001, PUN day *p* < 0.0001, glucose day *p* < 0.0001, and *n* = 30 cows per day.

**Figure 6 animals-13-03185-f006:**
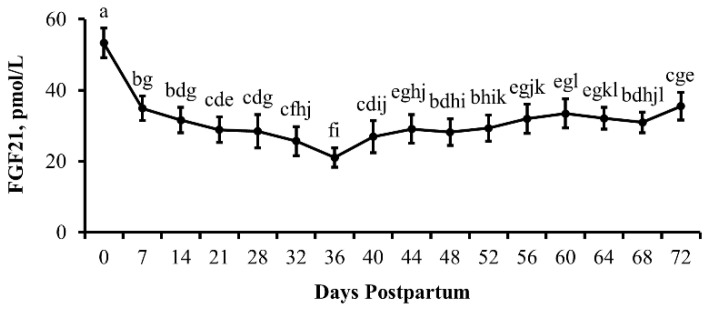
**Circulating concentrations of fibroblast growth factor 21 (FGF21) are elevated following parturition and during the period of peak lactation in beef cows:** Arithmetic means (±SEM) serum concentrations of FGF21 (pmol/L) in beef cows by day postpartum. Means with different lowercase letters denote differences between days. Day *p* < 0.0001; *n* = 46, 45, 47, 47, 47, 45, 46, 47, 47, 46, 47, 47, 47, 46, 46, and 46 cows per day for days 0, 7, 14, 21, 28, 32, 36, 40, 44, 48, 52, 56, 60, 64, 68, and 72, respectively.

**Figure 7 animals-13-03185-f007:**
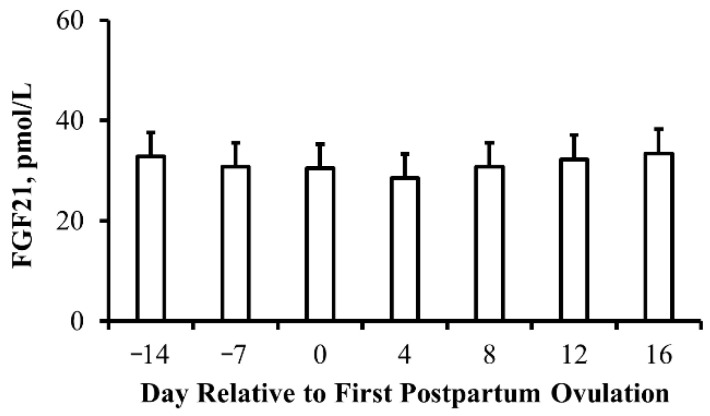
**Serum concentrations of fibroblast growth factor 21 (FGF21) fail to fall prior to the postpartum resumption of reproductive cyclicity in beef cows:** Least-squares means (±SEM) serum concentrations of FGF21 (pmol/L) relative to the first postpartum ovulation. Day *p* = 0.45 and *n* = 37, 40, 39, 35, 39, 34, and 34 cows on days −14, −7, 0, 4, 8, 12, and 16 relative to first postpartum ovulation.

**Figure 8 animals-13-03185-f008:**
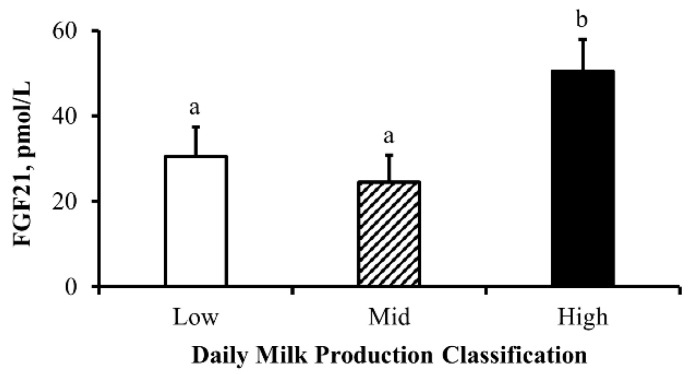
**Concentrations of fibroblast growth factor 21 (FGF21) differ by level of milk production in beef cows:** Least-squares means (+SEM) serum concentrations of FGF21 (pmol/L) on experimental day 60 in beef cows by daily milk production classification assigned on day 60. Individual daily milk production was ranked relative to 0.5 standard deviations below (Low; <10.59 kg; white bar) and above (High; >19.48 kg; dashed bar) the mean (Mid; 15.04 kg; black bar) daily milk production. Means with different lowercase letters denote differences between daily milk production classifications. Classification *p* = 0.03 and *n* = 15, 18, and 13 cows classified as Low, Mid, and High daily milk producers, respectively.

## Data Availability

Data will be made available upon reasonable request to the corresponding author.

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
