# Peer review of "Fibroblast Growth Factor 21 Has a Diverse Role in Energetic and Reproductive Physiological Functions of Female Beef Cattle"

_animals, 2023, doi:10.3390/ani13203185_

Round 1
Reviewer 1 Report
This manuscript reports several experiments dealing with the effects of several conditions and factors on FGF21 plasma concentrations in female beef cattle. Based on the fact that less is known about the function of FGF21, these studies add some new knowledge in this respect. The manuscript is largely well written and easy to read and follow.
There are only few comments on this manuscript:
One major hypothesis that FGF21 plays a role in female reproductivity. This hypothesis has been already raised in a recent review of the effects and the potential relevance of FGF21 in dairy cows published in Journal of Animal Science and Biotechnology 2021. This should be mentioned.
Figure 5: Regarding a relationship between FGF21 and PUN: Recent studies have shown that an overexpression of FGF21 is associated with an increase in autophagy and muscle loss, suggesting that FGF21 could play a role in muscle proteolysis which might be in close relationship to PUN levels (for literature see the review of FGF21 in dairy cows)
Figure 8: Daily milk production and FGF21: It is likely that energy balance is more relevant for FGF21 than milk production. In most cases, higher milk production is associated with a more pronounced negative energy balance – which as a result of higher NEFA plasma concentrations induces FGF21 expression due to activation of PPARalpha. There is however also a recent study showing that cows with high or low FGF21 expression do not differ in milk production (Animals 13, 231, 2023), suggesting that milk performance does not influence FGF21 expression per se.
Author Response
Dear Reviewer 1,
Thank you for your thoughtful review of our manuscript. Below we have addressed each of your comments.
Comment 1: One major hypothesis that FGF21 plays a role in female reproductivity. This hypothesis has been already raised in a recent review of the effects and the potential relevance of FGF21 in dairy cows published in Journal of Animal Science and Biotechnology 2021. This should be mentioned.
The authors thank the reviewer for this insight, but respectfully decline to include reference to a review article as it is better suited to reference the original work that is supported by data. Moreover, the present manuscript is in reference to beef cattle which respond considerably different to metabolic and physiologic stressors than dairy cattle, likely as a function of genetic selection practices.
Comment 2: Figure 5: Regarding a relationship between FGF21 and PUN: Recent studies have shown that an overexpression of FGF21 is associated with an increase in autophagy and muscle loss, suggesting that FGF21 could play a role in muscle proteolysis which might be in close relationship to PUN levels (for literature see the review of FGF21 in dairy cows)
This relationship has now been included in Lines 473 – 475.
Comment 3: Figure 8: Daily milk production and FGF21: It is likely that energy balance is more relevant for FGF21 than milk production. In most cases, higher milk production is associated with a more pronounced negative energy balance – which as a result of higher NEFA plasma concentrations induces FGF21 expression due to activation of PPARalpha. There is however also a recent study showing that cows with high or low FGF21 expression do not differ in milk production (Animals 13, 231, 2023), suggesting that milk performance does not influence FGF21 expression per se.
The reference suggested pertains to hepatic expression of FGF21 in dairy cattle and is beyond the scope of the current manuscript that focuses on circulating metabolites as it pertains to beef cattle that failed to exhibit differences in growth performance.
Sincerely,
Jennifer
Reviewer 2 Report
This report describes profiles of FGF 21 along with puberty, dietary change, estrus cycle, parturition and lactation in beef cattle.
Authors evaluate mainly discussed on relation of the changes of circulating FGF21 and requirement of FGF21 for different energy expenditure. Source of FGF21 at different reproductive status may need to be discussed, such as placenta along with parturition.
Author Response
Dear Reviewer 2,
Thank you for your thoughtful review of our manuscript. Below we have addressed your comment.
Authors evaluate mainly discussed on relation of the changes of circulating FGF21 and requirement of FGF21 for different energy expenditure. Source of FGF21 at different reproductive status may need to be discussed, such as placenta along with parturition.
The authors greatly appreciate the reviewers comment and have now included within the manuscript a section discussing the potential role of placenta-derived FGF21 during the periparturient period (Lines 435 – 439).
Sincerely,
Jennifer
Reviewer 3 Report
Need to adjust the font size of the text in the following places:
L94-96: “comprised of 94 75% Red Angus and 25% MARC III composites (25% Red Angus, 25% Hereford, 95 25% Pinzgauer, 25% Red Poll”
L187-189: “On day 60, calves were isolated from their dams for 8 hours and the weight-suckle-weigh procedure [10] was performed in groups of 5 to 7 calves to estimate daily milk production of the dam. Individual ADG was ranked relative to 0.5 standard deviations above and below the targeted ADG”
L192: “for statistical analysis. Individual”
L359: “Individual d”
L367: review word “circulating”
Author Response
Dear Reviewer 3,
Thank you for your thoughtful review of our manuscript. Below we have addressed each of your comments.
Need to adjust the font size of the text in the following places:
L94-96: “comprised of 94 75% Red Angus and 25% MARC III composites (25% Red Angus, 25% Hereford, 95 25% Pinzgauer, 25% Red Poll”
Thank you, the authors have now corrected this error.
L187-189: “On day 60, calves were isolated from their dams for 8 hours and the weight-suckle-weigh procedure [10] was performed in groups of 5 to 7 calves to estimate daily milk production of the dam. Individual ADG was ranked relative to 0.5 standard deviations above and below the targeted ADG”
This error has now been corrected.
L192: “for statistical analysis. Individual”
This error has now been corrected.
L359: “Individual d”
This error has now been corrected.
L367: review word “circulating”
This error has now been corrected.
Sincerely,
Jennifer
Reviewer 4 Report
The MS entitled “Fibroblast growth factor 21…” by Prezotto et al investigated blood FGF21, NEFA, urea nitrogen, glucose, progesterone, backfat thickness and milk production, indicating that FGF21 has an expansive role in physiology of female beef cattle, including pubertal onset, adaptation to nutritional transition, body weight gain, biochemical indexes, and milk production, which provides a novel means for beef cattle producers to select parameters of financial interest.
General comment
The MS is basically well prepared and the content including 4 experiments is in rich for publication in Animals. However, there are some problems with statistical labeling that needs to be modified, and the languages need to be improved.
Specific comment
1. Figure 1. the labels should be a, ac, abc, abc, bc, ac, b for seven timepoints?
Other figures also need to be checked and labelled correctly, especially for figure 6 which is too complicated.
2. It’s better for the authors to analyze the correlation coefficient of FGF21 with the detected indexes to confirm the potential role of FGF21 in the physiology of female beef cattle, like FGF21 and the ADG over the course of the 56-day feeding period in L262
3. The statics analysis should be integrated into another subtitle as 2.4
4. Languages need to be reorganized, For example:
L41 “Change in feed quality and availability alter the” should be changed into “Changes in feed quality and availability alter the”
L50 “characteristic of a marked periparturient increase in circulating” should be changed into “with a marked periparturient increase in circulating”
L55 and other metabolic markers by others, what means?
“the relationship of FGF21 to parturition” should be changed into “the relationship L56 between FGF21 and parturition”
1. Languages need to be reorganized, For example:
L41 “Change in feed quality and availability alter the” should be changed into “Changes in feed quality and availability alter the”
L50 “characteristic of a marked periparturient increase in circulating” should be changed into “with a marked periparturient increase in circulating”
L55 and other metabolic markers by others, what means?
“the relationship of FGF21 to parturition” should be changed into “the relationship L56 between FGF21 and parturition”
Author Response
Dear Reviewer 4,
Thank you for your thoughtful review of our manuscript. Below we have addressed each of your specific comments.
- Figure 1. the labels should be a, ac, abc, abc, bc, ac, b for seven timepoints?
Edited as suggested.
Other figures also need to be checked and labelled correctly, especially for figure 6 which is too complicated.
Thank you. The authors have confirmed the superscripts denoting p-diff in all other figures and that the p-diff for Figure 6 have been presented in the most simple manner.
- It’s better for the authors to analyze the correlation coefficient of FGF21 with the detected indexes to confirm the potential role of FGF21 in the physiology of female beef cattle, like FGF21 and the ADG over the course of the 56-day feeding period in L262
Thank you for this comment, the authors agree and this is why the correlation coefficient has already been included within the manuscript as r-squared in lines 261 and 262.
- The statics analysis should be integrated into another subtitle as 2.4
Thank you for your suggestion, but as the statistical analyses are different by experiment the authors believe it is best separated by experiment.
- Languages need to be reorganized, For example:
L41 “Change in feed quality and availability alter the” should be changed into “Changes in feed quality and availability alter the”
Edited as suggested.
L50 “characteristic of a marked periparturient increase in circulating” should be changed into “with a marked periparturient increase in circulating”
Edited as suggested.
L55 and other metabolic markers by others, what means?
The authors mean that other markers of metabolites have also been assessed in the referenced manuscripts. However, they have not been listed as the number of metabolic markers is expansive and not directly relevant to the subject at hand.
“the relationship of FGF21 to parturition” should be changed into “the relationship L56 between FGF21 and parturition”
Edited as suggested.
Sincerely,
Jennifer